# Probabilistic Prototype Generation Network for Cross-Domain Few-Shot Semantic Segmentation

## Abstract

Cross-domain few-shot semantic segmentation (CD-FSS) aims to tackle the challenge of adapting models from labeled source domains to unseen target domains with novel classes and limited annotations. Existing methods predominantly rely on straightforward support-query feature matching, making them vulnerable to domain shifts and limiting their generalization. In contrast, vision foundation models (VFMs) based on Transformer architectures demonstrate exceptional cross-domain transferability by offering powerful off-the-shelf global contextual priors. To this end, we propose a novel probabilistic prototype generation network (PPGN), which integrates global contextual priors from VFMs to enhance prototype representation learning with probabilistic modeling for CD-FSS. Specifically, PPGN adopts a dual-encoder architecture that incorporates DINOv2's capability of global contextual modeling with conventional CNN-based local feature extraction, thus leading to more comprehensive visual representations. We first design a dynamic prototype generator (DPG), which exploits high-confidence response maps from both branches to guide the generation of discriminative query prototypes, mitigating the inherent support-query divergence. Next, we propose a mixed-probabilistic prototype generator (MPG) that performs probabilistic modeling on the hybrid prototype integrated from heterogeneous feature spaces to enhance prototype generalization. Finally, an adaptive prediction aggregator (APG) is leveraged to refine segmentation by recalibrating and integrating multi-stage predictions. Extensive experiments demonstrate that PPGN achieves state-of-the-art performance on four CD-FSS benchmarks.

## 1 Introduction

In recent years, semantic segmentation has achieved remarkable progress (Minaee et al., 2021), largely driven by the availability of large-scale pixel-level annotated datasets. However, creating such annotations is extremely time-consuming and labor-intensive, which poses a significant barrier to extensive applications. To mitigate this challenge, few-shot semantic segmentation (FSS) (Dong & Xing, 2018) has emerged as a promising alternative, aiming to learn a model that can segment novel semantic classes trained with only a few annotated samples. Despite significant advances (Liu et al., 2020a; Xie et al., 2021a;b; Lu et al., 2021), the practicality of FSS remains limited by the critical assumption that the source and target domains share an identical data distribution. In reality, this assumption is often violated due to the prevalence of domain shifts, leading state-of-the-art FSS models to suffer substantial performance degradation when applied to target domains that differ markedly from the source (*e.g.*, transferring from PASCAL VOC (Everingham et al., 2010) to DeepGlobe (Demir et al., 2018)).

To address this limitation, FSS is extended to a new cross-domain few-shot segmentation task (CD-FSS). In this setting, models are trained on a source domain with abundant pixel-level annotations (*e.g.*, PASCAL VOC 2012 (Everingham et al., 2010)) and then adapted to a target domain that exhibits significant distribution shifts and novel class categories (*e.g.*, DeepGlobe (Demir et al., 2018)), with only a few annotated samples per class. CD-FSS has recently garnered increasing attention as a critical and challenging extension of conventional FSS, driving the development of numerous specialized approaches. For instance, some approaches (Fan et al., 2023; Nie et al., 2024;

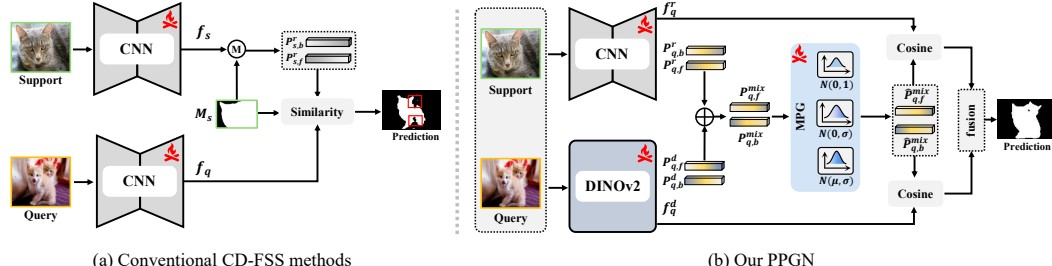

(a) Conventional CD-FSS methods              (b) Our PPGN

Figure 1: Comparison between existing CD-FSS methods and our approach. (a) Conventional CD-FSS methods typically adopt a single feature encoder and perform segmentation by directly matching query features with static support prototypes, resulting in limited discriminative capability and suboptimal performance. In contrast, (b) the proposed PPGN synergizes CNN-based local feature extraction with DINOv2's global contextual modeling and rich visual priors, enabling the learning of more comprehensive and robust visual representations. More importantly, the incorporation of probabilistic modeling over hybrid prototypes (*i.e.*, MPG) significantly enhances feature representation and generalization, thereby leading to more accurate segmentation.

Kong et al., 2024; Chen et al., 2024b) adopt fine-tuning strategies to rapidly adapt models pre-trained on source domains to target domains, effectively aligning with the target domain's feature distribution. Moreover, some recent works (He et al., 2024; Peng et al., 2025) incorporate foundational vision models into CD-FSS tasks to achieve global semantic consistency and improve feature representation learning. More recently, several methods (Tong et al., 2024; Liu et al., 2025b) employ the fast Fourier transform to convert spatial features into frequency domains, where reorganizing frequency and amplitude components helps the extraction of domain-invariant information for more accurate cross-domain transfer.

Although existing methods have shown promising results, several critical challenges remain unaddressed. First, most CD-FSS approaches rely on a single feature extractor, typically either a convolutional neural network (CNN) or a transformer, which is insufficient for mining the comprehensive visual semantics necessary for robust prototype representations. Second, inherent intra-class variations between support and query samples often cause feature mismatches, leading to error propagation during direct prototype-based matching and consequent performance degradation. Most critically, conventional static prototypes exhibit limited discriminative capacity, rendering them inadequate for accurately capturing category-specific features. As shown in Fig. 1 (a), traditional CD-FSS methods often produce erroneous predictions, such as misclassifying foreground regions (*e.g.*, *body of cats*) as background (highlighted by the red box).

To address the above issues, we propose a probabilistic prototype generation network (PPGN) for the CD-FSS task. The proposed method pioneers the integration of semantically rich visual priors from vision foundation models and introduces probabilistic modeling over prototype features, generating more discriminative and robust prototype representations, thereby significantly improving segmentation performance in CD-FSS. Specifically, PPGN synergistically combines the local feature extraction of ResNet50 (He et al., 2016) with rich global contextual visual priors of DINOv2 (Oquab et al., 2023) in a dual-branch feature encoder, capturing comprehensive and complementary visual representations of the object of interest. More importantly, to address the intrinsic feature distribution divergence between support and query images, we design a dynamic prototype generator (DPG), which leverages high-confidence maps co-predicted by ResNet-50 and DINOv2 to dynamically generate more discriminative query prototypes under the guidance of support prototypes. In addition, considering conventional single static prototypes often fail to accurately capture intraclass variability, we design a mixed-probabilistic prototype generator (MPG), which applies probabilistic modeling to hybrid prototypes integrated from heterogeneous feature spaces, thereby enhancing prototype robustness and generalization. Finally, we introduce an adaptive prediction aggregator (APG) based on a prototype recalibration strategy to improve prediction performance, which adaptively consolidates multi-stage predictions to refine the final segmentation output. As illustrated in Fig. 1 (b), PPGN is capable of learning more discriminative and generalizable foreground prototypes, enabling precise delineation of target objects (*e.g.*, accurate segmentation of the *cats* foreground region). In summary, our contributions can be summarized as follows:

- To the best of our knowledge, our proposed PPGN is the first framework that systematically incorporates CNN's local feature extraction with global context prior features of DINOv2 to generate discriminative and robust prototypes under probabilistic learning, which establishes a new paradigm for future research in the CD-FSS community.

- We design a dynamic prototype generator (DPG) that leverages high-confidence response maps from dual branches to guide query prototype generation, significantly enhancing prototype discriminability.

- We propose a mixed-probabilistic prototype generator (MPG) that effectively integrates category prototypes from heterogeneous feature spaces and incorporates probabilistic modeling through non-deterministic parameterization, thereby improving prototype generalization.

- Building on a prototype recalibration strategy, we introduce an adaptive prediction aggregator (APG) that effectively combines multi-stage predictions through adaptive fusion, thereby enhancing overall model performance.

## 2 RELATED WORK

**Few-Shot Semantic Segmentation.**    FSS aims to segment objects of interest using only a few labeled samples, which can be broadly categorized into two types: prototype-based methods and parameterized methods. Prototype-based methods (Liu et al., 2020b; Li et al., 2021) extract representative prototypes from limited samples, classifying new instances by their similarity to each class prototype for precise pixel-level segmentation. More recently, some studies (Zhang et al., 2019; 2021) highlight that the use of a single prototype typically fails to adequately capture complete object characteristics. To overcome this issue, recent methods, *i.e.*, PRMMS (Tian et al., 2020) and ASGNet (Li et al., 2021), explore multi-prototype strategies to achieve holistic object representation. On the other hand, approaches (Tian et al., 2020; Min et al., 2021; Lang et al., 2022) based on learnable parameters generally adopt an encoder-feature processor-decoder architecture. The model parameters are adaptively optimized during training to capture inter-sample similarity patterns. Although effective for novel class segmentation within a single domain, these methods face generalization challenges across domains due to substantial data distribution differences. Overcoming this domain gap with limited annotations remains a significant challenge.

**Cross-Domain Semantic Segmentation.**    Current research on cross-domain semantic segmentation primarily encompasses two approaches: domain adaptation (DA) and domain generalization (DG). DA typically fine-tunes a source-trained model on the target domain data. Current approaches focus on adversarial domain alignment (Kang et al., 2018), unsupervised learning with pseudo-labels (Yuan et al., 2024), and combining adversarial adaptation with self-training or pixel-level adaptation (Du et al., 2024). In contrast, DG assumes no access to target domain data during training. Existing methods (Dou et al., 2019; Min et al., 2021; Peng et al., 2022; Zhao et al., 2024) can be categorized into two paradigms based on representation learning: domain invariance and feature disentanglement. The former achieves generalization through invariant risk minimization, kernel-based methods, explicit feature alignment, and adversarial domain learning. The latter enhances generalization by decomposing features into domain-shared and domain-specific components.

**Cross-Domain Few-Shot Semantic Segmentation.**    Traditional FSS models struggle in CD-FSS due to significant domain shifts and disjoint label spaces between the source and target domains. To mitigate this domain gap, recent advances have introduced several advanced approaches. For instance, PATNet (Lei et al., 2022) introduces the first standardized CD-FSS framework, employing a pyramidal adaptation module to transform domain-specific features into domain-agnostic representations, effectively resolving cross-domain feature discrepancies. DSFM (Kong et al., 2024) employs a parameter-free grouped style modulation (GSM) layer to generate diverse domain styles, improving model generalization, and adopts a dual-branch fusion (DBF) strategy to enhance discriminative capability and adaptability in target domains. GPRN (Peng et al., 2025) employs a SAM-aware prompt initialization (SPI) module to transform SAM-generated masks into semantically rich visual prompts, while utilizing a graph prompt reasoning (GPR) module to construct relational graphs among prompts, thereby achieving global semantic consistency and enhancing feature representation learning. DR-Adapter (Su et al., 2024) simulates diverse target domain features through local-global

style perturbation and aligns them to the source domain space via cyclic alignment loss, enhancing robustness to domain shifts while reducing overfitting.

Unlike the above methods, our approach first establishes a dual-encoder architecture that synergistically combines ResNet50's local feature extraction with DINOv2's global context modeling, thereby capturing effective cross-domain representations. Furthermore, we utilize high-confidence response maps from both branches to produce more discriminative query prototypes under the guidance of support prototypes. To further enhance generalization, we introduce probabilistic hybrid prototypes via non-deterministic modeling, which explicitly captures intra-category relationships. Finally, based on a prototype recalibration strategy, we develop an adaptive prediction aggregator that effectively combines multi-stage predictions through adaptive fusion, achieving superior segmentation accuracy.

## 3 METHOD

### 3.1 PROBLEM DEFINITION

In the CD-FSS task, the model is first trained in the source domain ($\{\mathcal{X}_s, \mathcal{Y}_s\} \in \mathcal{D}_{source}$) and then tested and evaluated in the target domain ($\{\mathcal{X}_t, \mathcal{Y}_t\} \in \mathcal{D}_{target}$). Here, $\mathcal{X}_s$ and $\mathcal{X}_t$ denote the data distributions, while $\mathcal{Y}_s$ and $\mathcal{Y}_t$ are the corresponding label spaces. Note that $\mathcal{X}_s$ and $\mathcal{X}_t$ have different data distributions, and $\mathcal{Y}_s$ and $\mathcal{Y}_t$ share no intersection, *i.e.*, $\mathcal{X}_s \neq \mathcal{X}_t$, $\mathcal{Y}_s \cap \mathcal{Y}_t = \emptyset$. Specifically, in an $N$-way $K$-shot setting, both the training set ($\mathcal{D}_{train}$) and the testing set ($\mathcal{D}_{test}$) consist of multiple episodes, each containing a support set ($\mathcal{S} = \{I_i^s, M_i^s\}_{i=1}^{N \times K}$) and a query set ($\mathcal{Q} = \{I_i^q, M_i^q\}_{i=1}^{Q}$), where $I_i^s \in \mathbb{R}^{H \times W \times 3}$ and $I_i^q \in \mathbb{R}^{H \times W \times 3}$ denote RGB images and $M_i^s$ and $M_i^q$ represent their corresponding binary masks.

### 3.2 OVERVIEW

As illustrated in Fig. 2, our PPGN comprises three key components: (a) dual-branch feature encoders based on CNN and Transformer architectures, (b) dynamic mixed-probabilistic prototype generator (DMPG), and (c) a prototype recalibration generator (PRG). Specifically, we employ ResNet50 (He et al., 2016) ($E^r$) and DINOv2 (Oquab et al., 2023) ($E^d$) as dual-branch feature encoders to extract high-level semantic features from input support and query samples, *i.e.*, $\{f_s^r, f_q^r\}$ = $E^r(I_s, I_q)$ and $\{f_s^d, f_q^d\} = E^d(I_s, I_q)$. To address inherent intra-class variations, we propose the dynamic mixed-probabilistic prototype generator (DMPG), which contains two key components: 1) a dynamic prototype generator (DPG) that generates discriminative query prototypes (*i.e.*, $\{P_{q,f}^r, P_{q,b}^r\}$ and $\{P_{q,f}^d, P_{q,b}^d\}$) by leveraging high-confidence response maps from both ResNet50 and DINOv2 predictions (*i.e.*, $\{M_r^i, M_d^i\}$, $i = \{1, 2\}$), guided by support prototypes; and 2) a mixed-probabilistic prototype generator (MPG) that enhances prototype generalization through adaptive cross-space prototype fusion with probabilistic non-deterministic parameterization. Additionally, to further enhance model performance, we propose a prototype recalibration generator (PRG) with an adaptive prediction aggregator (APG) module that effectively integrates multi-stage predictions through learned complementary weighting, yielding significant performance gains.

### 3.3 DYNAMIC MIXED-PROBABILISTIC PROTOTYPE GENERATOR

**Dynamic Prototype Generator.** As discussed, inherent intra-class variations between support and query samples can propagate erroneous during direct prototype–feature similarity computation, ultimately degrading model performance. To address this issue, we specifically design a dynamic prototype generator (DPG) that enhances discriminative capability by guiding the generation of the query prototype through high-confidence dual-branch response maps, as shown in Fig. 2 (a). Specifically, for the visual features extracted from both ResNet50 (*i.e.*, $f_s^r$ and $f_q^r$) and DINOv2 (*i.e.*, $f_s^d$ and $f_q^d$) branches, we first perform bilinear interpolation and $1 \times 1$ convolution on both visual features, respectively, for dimension alignment. Then, we obtain initial prototype representations of support images through masked average pooling (MAP), which can be denoted as:

$$\{P_{s,f}^r, P_{s,b}^r\} = \mathcal{F}_{MAP}(f_s^r, M_s), \{P_{s,f}^d, P_{s,b}^d\} = \mathcal{F}_{MAP}(f_s^d, M_s), \tag{1}$$

where $\mathcal{F}_{MAP}(\cdot)$ denotes the masked average pooling operation. $P_{s,f}^r$ and $P_{s,b}^r$ represent the prototypes in the foreground and background obtained from the ResNet50 branch, while $P_{s,f}^d$ and $P_{s,b}^d$ correspond

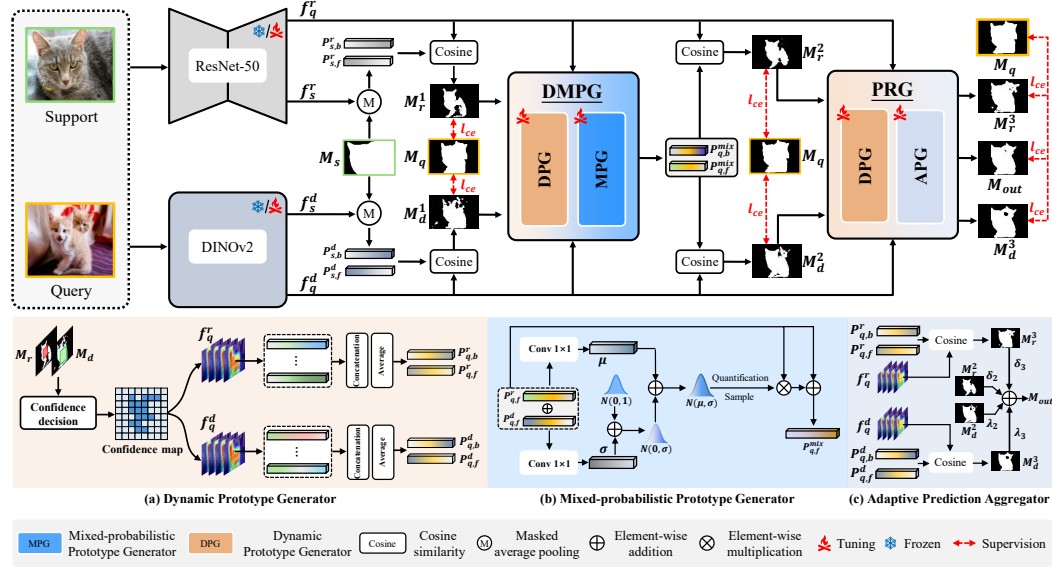

Figure 2: Architecture of the proposed PPGN. PPGN pioneers the integration of ResNet50's local feature extraction with DINOv2's global context representations, thereby facilitating comprehensive and complementary visual feature learning. Subsequently, we design a dynamic mixed-probabilistic prototype generator (DMPG), which comprises a dynamic prototype generator (DPG) and a mixed-probabilistic prototype generator (MPG). The DMPG produces discriminative query prototypes using high-confidence maps from both branches, while boosting generalizability through probabilistic hybrid prototypes (§3.3). Finally, we introduce a prototype recalibration generator (PRG), consisting of DPG and an adaptive prediction aggregator (APG), to adaptively aggregate multi-stage predictions, ultimately improving overall segmentation performance (§3.4).

to those derived from the DINOv2 branch. $M_s$ indicates the ground-truth mask of the support image. Subsequently, we compute the cosine similarity between the support prototypes and query features to generate initial query predictions, $i.e.$, $M_r^1$ and $M_d^1$, which can be expressed as:

$$M_r^1 = \mathcal{F}_{cosine}(f_q^r, P_{s,f}^r, P_{s,b}^r), \ M_d^1 = \mathcal{F}_{cosine}(f_q^d, P_{s,f}^d, P_{s,b}^d), \tag{2}$$

where $\mathcal{F}_{cosine}(\cdot)$ indicates the cosine similarity operation. For the initial query predictions $M_r^1$ and $M_d^1$ obtained from the ResNet50 and DINOv2 branches, respectively, we can obtain a confidence map through confidence computation. Specifically, channel-wise softmax normalization is first applied to map the values of each spatial location's pixel to the interval (0, 1). Here, the pixel values in each channel represent the confidence scores for the foreground/background classification. To generate more discriminative query prototypes (taking foreground prototype calculation as an example), we introduce a learnable foreground threshold $\tau$ (initialized to 0.5). Positions that simultaneously exceed $\tau$ in both $M_r^1$ and $M_d^1$ are identified as high-confidence response points. The corresponding feature vectors of $f_q^r$ and $f_q^d$ at these locations serve as components to construct the foreground prototype. After traversing all spatial coordinates in $M_r^1$, we aggregate the candidate set of foreground vectors to generate branch-specific foreground prototypes. This process is formulated as:

$$\Omega^{r/d} = \left\{ f_q^{r/d}(x,y) \mid \forall (x,y), M_r^1(x,y) \geq \tau \cap M_d^1(x,y) \geq \tau \right\}, \tag{3}$$

where $(x,y)$ is spatial coordinates, $\Omega^r$ and $\Omega^d$ denote the sets of high-confidence foreground vectors selected from $f_q^r$ and $f_q^d$, respectively. We then construct the foreground prototype for the query target by concatenating and averaging the high-confidence foreground vectors, denoted as:

$$P_{q,f}^{r/d} = \mathcal{F}_{avg}(\mathcal{F}_{cat}(\omega_1, \omega_2, \ldots, \omega_n)), \ \{\omega_1, \omega_2, \ldots, \omega_n\} \in \Omega^{r/d}, \tag{4}$$

where $\mathcal{F}_{cat}(\cdot)$ and $\mathcal{F}_{avg}(\cdot)$ denote channel-wise concatenation and average operations, respectively. Additionally, it is worth noting that we can obtain the background prototype of the query in the same manner, $i.e.$, $P_{q,b}^{r/d}$. Finally, we can obtain dual prototype representations for each query through the ResNet50 and DINOv2 branches, $i.e.$, $\{P_{q,b}^r, P_{q,f}^r\}$ and $\{P_{q,b}^d, P_{q,f}^d\}$.

**Mixed-probabilistic Prototype Generator.** Conventional CD-FSS methods typically employ a single static prototype to represent the target category in query images. However, such a simplistic and fixed representation often fails in complex scenarios. To address this limitation, we propose a mixed-probabilistic prototype generator (MPG), as illustrated in Fig. 2 (b), to enhance prototype generalization through non-deterministic modeling of hybrid prototypes with probabilistic parameterization. Specifically, we adaptively fuse the query prototypes captured from both DINOv2 and ResNet50 branches using learnable parameters (taking the mixed foreground prototype as an example). This process generates category prototypes that effectively integrate information from different feature spaces, expressed as:

$$P_{q,f}^{mix} = \alpha \cdot P_{q,f}^{r} + \beta \cdot P_{q,f}^{d}, \tag{5}$$

where $\alpha$ and $\beta$ represent learnable parameters. We further enhance the discriminative ability of the query prototype by performing probabilistic modeling on the mixed foreground prototype, *i.e.*, $P_{q,f}^{mix}$, through parameterized probability distributions. Specifically, we adopt two learnable $1\times1$ convolutional layers to learn the mean (termed as $\mu$) and variance (denoted as $\sigma$) of the mixed foreground prototype, expressed as:

$$\mu = \mathcal{F}_{conv_\mu}(P_{q,f}^{mix}), \sigma = \mathcal{F}_{conv_\sigma}(P_{q,f}^{mix}), \tag{6}$$

where $\mathcal{F}_{conv_\mu}(\cdot)$ and $\mathcal{F}_{conv_\sigma}(\cdot)$ denote learnable $1\times1$ convolutional layers. Besides, to facilitate model learning, we introduce an external standard Gaussian distribution, *i.e.*, $\varepsilon \sim \mathcal{N}(0,1)$, to dynamically weight the distribution parameters of the mixed foreground prototype. Simultaneously, to mitigate feature drift in category prototypes, we employ reparameterization and multi-sampling strategies to enhance their discriminative capabilities. After $N$ sampling iterations, we obtain the following sampled set:

$$P = \{p_i = \mu + \varepsilon_i \sigma \mid i = 1, 2, ..., N\}. \tag{7}$$

Then, we stack the $N$ sampled prototypes (*i.e.*, $p_i$) along the 0-th dimension and compute their variances to quantify the uncertainty (termed as $U$) in prototype representations. This uncertainty modeling process can be denoted as:

$$U = \mathcal{F}_{var}(\mathcal{F}_{stack}(P, \ dim = 0), dim = 0), \tag{8}$$

where $\mathcal{F}_{stack}(\cdot)$ and $\mathcal{F}_{var}(\cdot)$ represent stack and variance calculation operation, respectively. To enhance the deterministic characteristics of category prototypes while filtering stochastic noise, we first invert the captured uncertainty features and perform element-wise multiplication with the original mixed prototype. This operation selectively amplifies discriminative patterns. Furthermore, to preserve the integrity of initial prototype features and prevent information loss, we employ residual connections to fuse the refined prototype with the original mixed prototype, yielding the final mixed probabilistically-enhanced foreground prototype, *i.e.*, $\hat{P}_{q,f}^{mix}$. This refinement process is expressed as:

$$\hat{P}_{q,f}^{mix} = (1 - U) \otimes P_{q,f}^{mix} \oplus P_{q,f}^{mix}, \tag{9}$$

where $\otimes$ and $\oplus$ are element-wise multiplication and addition, respectively. Following the same procedure, we can compute the mixed probabilistically enhanced background prototype, *i.e.*, $\hat{P}_{q,b}^{mix}$.

## 3.4 PROTOTYPE RECALIBRATION GENERATOR

Prototypes and predictions exhibit a mutually reinforcing relationship, where iterative prediction–prototype–prediction co-calibration progressively enhances segmentation accuracy. To this end, we propose a prototype recalibration generator (PRG) that employs DPG to recalibrate and refine query representations, coupled with an adaptive prediction aggregator (APG) to adaptively integrate multi-stage predictions for performance enhancement, shown in Fig. 2 (c). Specifically, we compute the second-round query predictions by measuring the similarity between the enhanced mixed probabilistic prototypes (*i.e.*, $\hat{P}_{q,b}^{mix}$ and $\hat{P}_{q,f}^{mix}$) and the query features extracted from both encoder branches, which can be expressed as:

$$M_r^2 = \mathcal{F}_{cosine}(f_q^r, \hat{P}_{q,f}^{mix}, \hat{P}_{q,b}^{mix}), \ M_d^2 = \mathcal{F}_{cosine}(f_q^d, \hat{P}_{q,f}^{mix}, \hat{P}_{q,b}^{mix}). \tag{10}$$

Subsequently, DPG is applied to derive enhanced prototype representations from the query features and updated query predictions. Next, APG is used to first compute the similarity of the enhanced prototypes with the original query features, thus obtaining the third-stage segmentation predictions,

*i.e.*, $M_r^3$ and $M_d^3$. Notably, unlike the DMPG, these query prototypes exhibit inherently high reliability and thus do not require additional probabilistic modeling. This operation can be defined as:

$$M_r^3 = \mathcal{F}_{cosine}(f_q^r, \mathcal{F}_{DPG}(f_q^r, M_r^2, M_d^2)), \ M_d^3 = \mathcal{F}_{cosine}(f_q^d, \mathcal{F}_{DPG}(f_q^d, M_r^2, M_d^2)), \quad (11)$$

where $\mathcal{F}_{DPG}(\cdot)$ represents the DPG operation. To further enhance prediction accuracy, an adaptive aggregator is introduced to adaptively combine multi-stage predictions from ResNet50 and DINOv2 branches through learnable weighting factors, denoted as:

$$M_{out} = \sum_{i=2}^{3} (\delta_i \cdot M_r^i + \lambda_i \cdot M_d^i), \quad (12)$$

where $\delta_i$ and $\lambda_i$ ($i \in \{2, 3\}$) denote learnable parameters.

### 3.5 Loss Function

During both training and fine-tuning phases, we employ cross-entropy loss ($\mathcal{L}_{ce}$) to supervise the segmentation predictions from all stages of both DINOv2 and ResNet50 branches, *i.e.*, $M_r^1 \sim M_r^3$, $M_d^1 \sim M_d^3$, and $M_{out}$, thereby optimizing model parameters. Furthermore, for the $\mu$ and $\sigma$ of query prototypes obtained via $1\times1$ convolutions in the MPG module, we impose Kullback-Leibler divergence (Kingma & Welling, 2013) regularization using a standard normal distribution as prior. Let $M_q$ denote the ground-truth of the query. Therefore, the total loss function is calculated as follows:

$$\mathcal{L}_{total} = \mathcal{L}_{ce}(M_{out}, M_q) + \sum_{i=1}^{3} (\mathcal{L}_{ce}(M_r^i, M_q) + \mathcal{L}_{ce}(M_d^i, M_q)) + \mathcal{D}(\mathcal{N}(\mu, \sigma) || \mathcal{N}(0, 1)). \quad (13)$$

## 4 Experiment

### 4.1 Benchmarks

Following the PATNet (Lei et al., 2022) experimental setup, we utilize the SBD-augmented PASCAL VOC 2012 (Everingham et al., 2010) dataset, which contains 20 common object categories, as our source domain for model training. Subsequently, we fine-tune and evaluate our model on four widely used benchmark datasets, including DeepGlobe (Demir et al., 2018), ISIC2018 (Tschandl et al., 2018), Chest X-ray (Candemir et al., 2013), and FSS1000 (Li et al., 2020). For more details about the test dataset, please refer to **Appendix A.2**. For performance evaluation, we adopt the mean Intersection over Union (mIoU) metric and report the average results of five independent trials. Each trial uses a unique random seed to ensure a comprehensive and robust evaluation. Additionally, we thoroughly test our model's performance under both 1-way 1-shot and 1-way 5-shot settings to provide a complete understanding of its capabilities and limitations. For more details about the implementation and source code, please refer to **Appendix A.3** and **Appendix A.1**.

### 4.2 Comparison with State-of-the-art Methods

**Quantitative Results.** We compare our PPGN with state-of-the-art CD-FSS models, and the results are shown in Table 1. It is worth noting that all models are trained on the PASCAL VOC 2012 (Everingham et al., 2010) dataset and evaluated quantitatively using mIoU (%) as the primary metric across four standard datasets. To ensure a fair comparison, we categorize the models based on whether they are fine-tuned, as fine-tuning typically leads to significant performance improvements. As shown in Table 1, our PPGN achieves a remarkable performance advantage. Specifically, under the 1-shot and 5-shot settings, our PPGN reaches impressive accuracies of 73.47% and 74.35%, respectively, on the ISIC2018 (Tschandl et al., 2018) dataset. More importantly, the average mIoU across the four standard datasets fully reflects the model's comprehensive generalization ability across different domains. Our PPGN achieves average mIoU scores of **72.18%** and **76.16%** on these four datasets, demonstrating state-of-the-art performance. Moreover, compared to GPRN (Peng et al., 2025), which employs other foundation vision models (*i.e.*, SAM), our PPGN outperforms it by ↑0.48% (1-shot) and ↑0.86% (5-shot) in average mIoU across four benchmark datasets. Additionally, compared to another outstanding model (*e.g.*, IFANet (Nie et al., 2024)), our PPGN achieves

Table 1: Quantitative comparison of the proposed method and state-of-the-art CD-FSS approaches on four benchmarks, *i.e.*, Deepglobe (Demir et al., 2018), ISIC2018 (Tschandl et al., 2018), Chest X-ray (Candemir et al., 2013) and FSS1000 (Li et al., 2020), evaluated by mIoU (%). The top two performances are highlighted in bold and underlined, respectively.

| Method | DeepGlobe | | ISIC2018 | | Chest X-ray | | FSS1000 | | Average | |
|---|---|---|---|---|---|---|---|---|---|---|
| | 1-shot | 5-shot | 1-shot | 5-shot | 1-shot | 5-shot | 1-shot | 5-shot | 1-shot | 5-shot |
| Methods without Fine-tuning Phase | | | | | | | | | | |
| AMP (Siam et al., 2019) [ICCV2019] | 37.64 | 40.63 | 28.41 | 30.49 | 51.27 | 53.07 | 57.23 | 59.22 | 43.63 | 45.82 |
| RestNet (Huang et al., 2023) [BMVC2023] | – | – | 42.25 | 51.10 | 70.43 | 73.69 | 81.53 | 84.89 | – | – |
| PMNet (Chen et al., 2024a) [WACV2024] | 37.10 | 41.60 | 51.20 | 54.50 | 70.40 | 74.00 | 84.60 | 86.30 | 60.83 | 64.10 |
| PerSAM (Zhang et al., 2024) [ICLR2024] | 30.02 | 30.14 | 23.30 | 25.35 | 61.07 | 66.52 | 36.13 | 40.74 | 37.62 | 40.68 |
| APM-M (Tong et al., 2024) [NeurIPS2024] | 40.86 | 44.92 | 41.71 | 51.16 | 78.25 | 82.81 | 79.29 | 81.83 | 60.03 | 65.18 |
| ABCDFSS (Herzog, 2024) [CVPR2024] | 42.60 | 49.00 | 45.70 | 53.30 | 79.80 | 81.40 | 74.60 | 76.20 | 60.70 | 65.00 |
| DR-Adapter (Su et al., 2024) [CVPR2024] | 41.29 | 50.12 | 40.77 | 48.87 | 82.35 | 82.31 | 79.05 | 80.40 | 60.86 | 65.42 |
| APSeg (He et al., 2024) [CVPR2024] | 35.94 | 39.98 | 45.43 | 53.98 | 84.10 | 84.50 | 79.71 | 81.90 | 61.30 | 65.09 |
| TVGTANet (Liu et al., 2025a) [ACM MM2025] | 42.04 | 50.67 | 47.21 | 58.75 | 84.58 | 87.27 | 78.32 | 81.44 | 63.04 | 69.53 |
| ISA (Fan et al., 2025) [ICCV2025] | 44.32 | 52.73 | 37.21 | 56.10 | 83.42 | 86.28 | 78.76 | 86.03 | 60.92 | 70.29 |
| LoEC (Liu et al., 2025b) [CVPR2025] | 44.10 | 49.67 | 38.21 | 47.04 | 81.02 | 82.73 | 78.51 | 80.60 | 60.46 | 65.01 |
| Methods with Fine-tuning Phase | | | | | | | | | | |
| PATNet (Lei et al., 2022) [ECCV2022] | 37.89 | 42.97 | 41.16 | 53.58 | 66.61 | 70.20 | 78.59 | 81.23 | 56.06 | 61.99 |
| DARNet (Fan et al., 2023) [Arxiv2023] | 44.61 | 54.05 | 47.81 | 60.52 | 81.22 | 89.73 | 76.41 | 83.24 | 62.51 | 71.89 |
| DMTNet (Chen et al., 2024b) [IJCAI2024] | 40.14 | 51.17 | 43.55 | 52.30 | 73.74 | 77.30 | 81.52 | 83.28 | 59.74 | 66.01 |
| DFSM (Kong et al., 2024) [ACM MM2024] | 40.99 | 52.69 | 57.02 | 64.77 | 91.49 | 92.90 | 85.44 | 90.24 | 68.74 | 75.15 |
| IFANet (Nie et al., 2024) [CVPR2024] | 50.60 | 58.80 | 66.30 | 69.80 | 74.00 | 74.60 | 80.10 | 82.40 | 67.80 | 71.40 |
| GPRN (Peng et al., 2025) [AAAI2025] | 51.70 | 59.30 | 66.80 | 72.20 | 87.00 | 87.10 | 81.10 | 82.60 | 71.70 | 75.30 |
| DFN (Tong et al., 2025a) [ICML2025] | 39.45 | 47.67 | 50.36 | 58.53 | 83.18 | 87.14 | 82.97 | 85.72 | 63.99 | 69.77 |
| SDRC (Tong et al., 2025b) [ICML2025] | 43.15 | 46.83 | 46.57 | 55.02 | 82.86 | 84.79 | 80.31 | 82.55 | 63.22 | 67.30 |
| DATO (Li et al., 2025) [CVPR2025] | 51.10 | 59.30 | 68.76 | 70.30 | 79.58 | 81.07 | 81.79 | 84.61 | 70.31 | 73.82 |
| **PPGN(Ours)** | 49.83 | **60.51** | **73.47** | **74.35** | 78.14 | 81.99 | **87.28** | 87.79 | **72.18** | **76.16** |

performance gains of ↑7.17%, ↑4.14%, and ↑7.18% under the 1-shot setting, and ↑4.55%, ↑7.39%, and ↑5.39% under the 5-shot setting on the ISIC2018 (Tschandl et al., 2018), Chest X-ray (Candemir et al., 2013), and FSS1000 (Li et al., 2020) datasets, respectively. The average improvements across the four standard datasets reach ↑4.38% and ↑4.76% for 1-shot and 5-shot scenarios, respectively.

**Qualitative Results.** We further visualize the prediction results of our PPGN across four benchmark datasets, as shown in Fig. 3 (results on DeepGlobe (Demir et al., 2018), ISIC2018 (Tschandl et al., 2018), Chest X-ray (Candemir et al., 2013), and FSS1000 (Li et al., 2020) datasets from top to bottom, respectively). Critically, these target-domain test categories not only exhibit distinct feature distributions from the source-domain data but are entirely absent during model training. Remarkably, under these extremely challenging cross-domain conditions, our PPGN consistently delivers accurate foreground segmentation. As evidenced by the Chest X-ray (Candemir et al., 2013) segmentation results in Fig. 3 (third row), the model

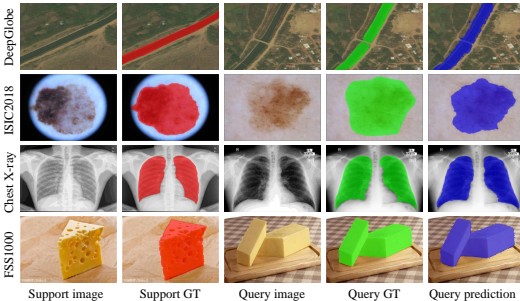

Figure 3: Visualization results on four benchmark datasets. The red, green, and blue regions denote the support masks, query masks, and query predictions, respectively.

successfully identifies precise lesion boundaries despite being trained solely on natural images, which exhibit substantial stylistic divergence from medical imaging. This compelling performance unequivocally demonstrates PPGN's exceptional generalization capacity and domain robustness. Further analysis of the generalization enhancement achieved by our proposed probabilistic mechanism under domain shift is provided in **Appendix A.4**.

### 4.3 ABLATION STUDY

We perform comprehensive ablation studies on the key components of PPGN to validate their effectiveness, conducting all comparative experiments under the most challenging 1-shot setting on the Chest X-ray (Candemir et al., 2013) dataset. Performance is evaluated by mIoU, with all results averaged over five random seed trials to ensure reliability. **More ablation studies are provided in Appendix A.5** .

**Component contribution analysis of DMPG and PRG.** To further validate the effectiveness of our proposed modules, *i.e.*, DMPG and PRG, we conduct progressive ablation experiments. First, we

establish pure ResNet50 and DINOv2 baselines (#3 in Table 2). Subsequent integration of the DMPG module yields a **+1.77%** performance improvement (#4 *vs.* #3). We analyze the potential reasons as follows: DPG produces more discriminative query prototypes, effectively addressing intra-class variations between support and query samples; MPG addresses the generalization limitations of conventional single prototypes by performing non-deterministic modeling of hybrid prototypes, thereby capturing probabilistic prototype representations that are more robust to domain variations. Further augmenting the model with the PRG achieves an additional **+2.94%** performance gain (#5 *vs.* #4 ). This improvement stems from PRG's adaptive prediction aggregator (APG), which strategically combines multi-stage segmentation outputs from dual branches through complementary prediction fusion, thereby fully exploiting cross-encoder synergies to enhance overall model capability.

**Effects of different encoders.** To rigorously validate the necessity of our dual-encoder architecture, we conduct comprehensive ablation experiments, with results detailed in Table 2. Specifically, we first remove all auxiliary components (including DMPG and PRG) and evaluate performance using individual encoders, *i.e.*, ResNet50 or DINOv2 alone. As shown in entries #1 and #2 of Table 2, the baseline models achieve 72.56% and 60.20% accuracy scores with standalone ResNet50 and DINOv2

Table 2: Ablation studies on key components in PPGN, including two distinct encoders (*e.g.*, ResNet50 and DINOv2), DMPG, and PRG. The best results are highlighted in bold.

| # | ResNet50 | DINOv2 | DMPG | PRG | mIoU (%) |
|---|---|---|---|---|---|
| 1 | ✓ | × | × | × | 72.56 |
| 2 | × | ✓ | × | × | 60.20 |
| 3 | ✓ | ✓ | × | × | 73.43 |
| 4 | ✓ | ✓ | ✓ | × | 75.20 |
| 5 | ✓ | ✓ | ✓ | ✓ | **78.14** |

encoders, respectively. Remarkably, when combining both encoders, the model attains a significantly higher accuracy of 73.43% (#3 in Table 2), delivering consistent improvements of +0.87% (#3 *vs.* #1) and +13.23% (#3 *vs.* #2), which conclusively demonstrates complementary advantages between the two encoding schemes.

**Component-wise analysis of DPG, MPG, and APG.** To further validate the contributions of finer-grained components in PPGN, including DPG, MPG, and APG. We conduct a series of ablation experiments, with quantitative results presented in Table 3. Specifically, we sequentially remove the DPG, MPG, and APG modules from the overall PPGN framework. The performance of model degrades by 1.07% (78.14% ↦ 77.07%), 3.11% (78.14% ↦ 75.03%), and 4.02% (78.14% ↦ 74.12%), respectively. We attribute these observations to the following: DPG removal forces the model to directly match query features

Table 3: Finer-grained ablation studies of the three key components in PPGN, *i.e.*, DPG, MPG and APG. The best results are highlighted in bold.

| # | DPG | MPG | APG | mIoU (%) |
|---|---|---|---|---|
| 1 | ✓ | ✓ | ✓ | **78.14** |
| 2 | × | ✓ | ✓ | 77.07 |
| 3 | ✓ | × | ✓ | 75.03 |
| 4 | ✓ | ✓ | × | 74.12 |

with support prototypes, amplifying segmentation errors due to inherent semantic discrepancies between support and query samples. The model's generalizability degrades significantly under MPG ablation, as being limited to deterministic prototype matching proves inadequate for handling complex scenarios where cluttered backgrounds demand more flexible representations. Removing APG reduces the fusion mechanism to a simple summation of dual-branch predictions, thereby losing both the adaptive weighting capability and the carefully designed complementary dynamics between encoders. In contrast, integrating all three modules achieves optimal accuracy, demonstrating their synergistic roles in the robust CD-FSS.

## 5 CONCLUSION

In this work, we propose a probabilistic prototype generation network (PPGN) for the CD-FSS task, which synergistically combines CNN's local feature extraction with the vision foundation model's global context prior knowledge to learn transferable cross-domain representations. Furthermore, we design a DPG, which produces more discriminative query prototypes through high-confidence response maps from dual-branch predictions, guided by support prototypes. Going beyond deterministic approaches, we develop a MPG that overcomes deterministic limitations via hybrid prototype modeling with probabilistic parameterization, significantly enhancing generalization. Finally, we introduce an APG to adaptively consolidate multi-stage segmentation results from dual branches, further improving the overall performance. Extensive experiments demonstrate that our method consistently outperforms existing approaches, establishing a new state-of-the-art benchmark.

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

## A APPENDIX

### A.1 SOURCE CODE

Code is available at `https://anonymous.4open.science/r/PPGN-4F8C`.

### A.2 TEST DATASET

We evaluate the cross-domain generalization capability of our segmentation model under varying domain gaps using four standard datasets from the CD-FSS task: DeepGlobe (Demir et al., 2018), ISIC2018 (Tschandl et al., 2018), Chest X-ray (Candemir et al., 2013), and FSS1000 (Li et al., 2020). These datasets cover satellite imagery, dermoscopic images of skin lesions, human lung X-ray images, and daily object images, respectively. The selected datasets exhibit category diversity, reflecting realistic scenarios for few-shot semantic segmentation. Specifically,

- **DeepGlobe** (Demir et al., 2018) is a satellite imagery dataset where each image is densely annotated at the pixel level with 7 categories: urban, agriculture, rangeland, forest, water, barren land, and unknown regions. The spatial resolution of the images is fixed at 2,448×2,448 pixels. To increase the number of test images and reduce their size, we split each image into 6 patches. After filtering out single-class images and those dominated by the "unknown" category, we retain 5,666 images (each 408×408 pixels) for final evaluation.
- **ISIC2018** (Tschandl et al., 2018) is a dermatological image dataset containing 2,596 lesion images, each with a single primary lesion. The original images have a spatial resolution of approximately 1,022×767 pixels. Following standard practice, we resize all images to 512×512 pixels for consistent processing.
- **Chest X-ray** (Candemir et al., 2013) is a medical imaging dataset for tuberculosis detection, comprising 566 X-ray images with an original resolution of 4,020×4,892 pixels. Given the large size of the raw images, we typically downsample them to 1,024×1,024 pixels for processing.
- **FSS1000** (Li et al., 2020) is a natural image dataset designed for few-shot segmentation, comprising 1,000 object categories with 10 samples per category. For our experiments, we adopt the official semantic segmentation split, where the test set contains 240 categories and 2,400 test images. Each test image features a single well-defined segmentation target to maintain task specificity. All test images maintain a standard resolution of 224×224 pixels.

### A.3 IMPLEMENTATION DETAILS

We employ the widely popular ResNet50 (He et al., 2016) as the backbone of our model. All experiments are conducted using PyTorch on a system equipped with a single NVIDIA RTX 4090 GPU. During the source domain training phase, we set the initial learning rate to $1\mathrm{e}^{-4}$ and the momentum to 0.9. Following the PATNet (Lei et al., 2022) paradigm, we standardize the input image resolution to $400 \times 400$. In the fine-tuning stage, we assign dataset-specific learning rates. For instance, the learning rates for the DeepGlobe (Demir et al., 2018), ISIC2018 (Tschandl et al., 2018), Chest X-ray (Candemir et al., 2013), and FSS1000 (Li et al., 2020) datasets are set to 0.002, 0.003, 0.0005, and 0.004, respectively. Each dataset undergoes 30 epochs of optimization, with the first 10 epochs dedicated to source domain training and the remaining 20 epochs for fine-tuning to enhance model performance. Furthermore, to augment the dataset and improve the robustness of the model, we adopt a series of data augmentation strategies provided by PyTorch, including horizontal flipping, vertical flipping, 90-degree rotation, and adjustments to brightness and hue. These techniques ensure the model effectively handles variations in input data.

### A.4 WHY THIS PROBABILISTIC MECHANISM ENHANCES GENERALIZATION UNDER DOMAIN SHIFT?

Probabilistic prototype modeling enhances the model's generalization ability by addressing the inherent uncertainty caused by limited data and intra-class variations, which is fundamentally based on Bayesian uncertainty modeling (Blundell et al., 2015; Kendall & Gal, 2017) to address domain shift challenges. Specifically, when $\mathcal{D}_{source}$ deviates from $\mathcal{D}_{target}$, deterministic prototypes (such

| $N$ | mIoU (%) |
|------|----------|
| 10 | 75.70 |
| 25 | 77.55 |
| **50** | **78.14** |
| 75 | 73.38 |
| 100 | 77.00 |
| 150 | 76.44 |

| $\tau$ | mIoU (%) |
|--------|----------|
| 0.5 | 77.94 |
| 0.6 | 68.11 |
| 0.7 | 50.86 |
| 0.8 | 45.84 |
| 0.9 | 49.47 |
| **learnable** | **78.14** |

| $\alpha$ | $\beta$ | mIoU (%) |
|----------|---------|----------|
| 0.1 | 0.9 | 77.91 |
| 0.3 | 0.7 | 78.09 |
| 0.5 | 0.5 | 78.07 |
| 0.7 | 0.3 | 77.59 |
| 0.9 | 0.1 | 76.10 |
| **learnable** | | **78.14** |

Table 4: Impact on the number of sampling iterations $N$ in the MPG.

Table 5: Ablation study on the learnable threshold $\tau$ in MPG.

Table 6: Influence of learnable parameters in MPG.

as mean vectors) often do not align with true class centroids. To this end, we model prototypes as Gaussian distributions $\mathcal{N}(\mu, \sigma^2)$, where the mean $\mu$ captures cross-domain invariant features, while the variance $\sigma^2$ quantifies uncertainty arising from domain-specific bias. Through qualitative analysis, our learned variance $\sigma^2$ could be regarded as the uncertainty score that measures the confidence of the embedded feature in belonging to the correct class. This process follows the principles of Bayesian deep learning (Kingma & Welling, 2013), as marginalizing over the noise distribution enhances the model's robustness to domain shifts.

Alternatively, from an intuitive perspective, the probabilistic modeling process can be viewed as an implicit data augmentation paradigm. The reparameterization technique performs multiple sampling of prototype distributions (as shown in Eq. 7), which can be viewed as applying smooth perturbations to category prototypes in the prototype space. This process effectively serves as implicit data augmentation, enhancing the model's robustness to input variations and consequently improving cross-domain generalization capability.

### A.5 MORE ABLATION STUDIES

**Different sampling iterations ($N$) in the MPG.** We conduct systematic experiments to evaluate different sampling iterations $N$ (10, 25, 50, 75, 100, 150) in the MPG module, with results shown in Table 4, where the model achieves optimal performance at 50 sampling iterations. We observe that insufficient sampling ($N < 50$) leads to significant errors in variance estimation and inaccurate hybrid prototype representations, while excessive sampling ($N > 50$) increases model complexity and disrupts the optimization process, causing performance degradation. As illustrated in Fig. 4 (a), the performance shows a clear trend: it improves monotonically as sampling iterations increase from 10 to 50, but degrades consistently beyond 50 samples. Based on this analysis, we set the default sampling iteration to 50 in PPGN as it provides the best balance between estimation accuracy and computational efficiency.

**Learnable threshold $\tau$ in MPG.** Investigating the role of threshold $\tau$ in MPG represents a crucial research focus. Through extensive comparative experiments (Table 5), we systematically evaluate the effectiveness of making $\tau$ a learnable parameter. When fixed to constant values (0.5, 0.6, 0.7, 0.8, 0.9), model performance progressively declines from 77.94% to 49.47%, as illustrated in Fig. 4 (b). In contrast, the learnable-$\tau$ configuration achieves optimal performance (78.14% mIoU on Chest X-ray dataset). We attribute this phenomenon to two key factors: (1) Overly large thresholds (*e.g.*, $\tau$ = 0.9) severely reduce valid feature vectors, compromising query prototype accuracy; and (2) our learnable-$\tau$ enables dynamic threshold adaptation across different domain distributions, generating more robust query prototypes through automatic confidence adjustment. Consequently, we implement $\tau$ as a learnable parameter in our PPGN.

**Effects of hyperparameter configurations in MPG.** In the MPG module, effectively fusing foreground and background prototypes from the ResNet50 and DINOv2 branches presents a critical challenge. To address this, we conduct a series of controlled experiments. Specifically, we evaluate different combinations of coefficients $\alpha$ and $\beta$, with quantitative results summarized in Table 6. The experimental results demonstrate that employing learnable coefficients $\alpha$ and $\beta$ yields optimal model accuracy (78.14% mIoU on Chest X-ray dataset). We posit that this adaptive fusion mechanism effectively combines ResNet-50's local inductive bias with DINOv2's global contextual awareness through learned optimal weighting, thereby generating a more generalizable prototype space compared to static aggregation. Thus, our PPGN adopts this dynamic weighting mechanism to aggregate both types of prototypes.

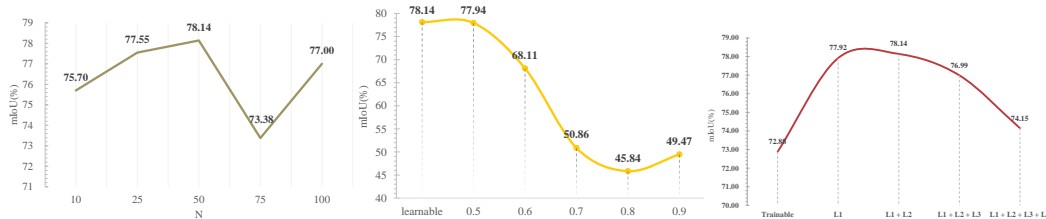

(a) Different numbers of sampling iterations ($N$).

(b) The impact of parameter $\tau$.

(c) Different frozen layers on ResNet50.

Figure 4: (a) Impact of varying sampling iterations in MPG. (b) Impact of threshold $\tau$ in the MPG. (c) Effects of varying frozen-layer configurations on model performance. "Trainable" denotes that all backbone layers remain learnable, while "L$i$" indicates that only the $i$-th layer is frozen, with other layers retaining trainable parameters.

**Impact on frozen layers in ResNet50.** To evaluate how freezing different layers in the ResNet50 (He et al., 2016) encoder affects model performance, we conduct a series of fine-grained ablation experiments on Chest X-ray dataset, with results presented in Table 7. Initially, we train the model with all layers learnable (Table 7 #1), then progressively freeze encoder layers from shallow to deep until the entire

Table 7: Impact on frozen layers in ResNet50.

| # | layer1 | layer2 | layer3 | layer4 | mIoU (%) |
|---|--------|--------|--------|--------|----------|
| 1 | ✓ | ✓ | ✓ | ✓ | 72.88 |
| 2 | × | ✓ | ✓ | ✓ | 77.92 |
| **3** | × | × | ✓ | ✓ | **78.14** |
| 4 | × | × | × | ✓ | 76.99 |
| 5 | × | × | × | × | 74.15 |

backbone is fixed, *i.e.*, #2 $\mapsto$ #5. Our findings show that PPGN performs best when layers $1 \sim 2$ are frozen while layers $3 \sim 4$ remain trainable. As depicted in Fig. 4 (c), model performance follows a bell-shaped curve — initially improving with partial freezing but declining when either too many or all layers are frozen. This trend suggests that fully training the ResNet50 (He et al., 2016) encoder introduces excessive learnable parameters, weakening cross-domain knowledge transfer, while completely freezing the backbone restricts feature adaptability, degrading representation quality. The optimal configuration (layers $1 \sim 2$ frozen, layers $3 \sim 4$ trainable) thus strikes a balance between preserving pretrained knowledge and enabling task-specific adaptation.

**Effects of hyperparameters in APG.** To further validate the role of adaptive fusion parameters in the APG module, we conduct an ablation study, with quantitative results summarized in Table 8. Specifically, when the learnable parameters (*e.g.*, $\delta_i$ and $\lambda_i$, $i \in \{2, 3\}$) are fixed to 1.0, the model's performance (mIoU on Chest X-ray datasets) decreases from 78.14% to 75.02%. We attribute this performance drop to the fact that adaptive fusion dynamically aggregates

Table 8: Effects of learnable parameters in APG.

| $\{\delta_2, \delta_3, \lambda_2, \lambda_3\}$ | mIoU (%) |
|---|---|
| fixed | 75.02 |
| **learnable** | **78.14** |

multi-stage segmentation results, effectively leveraging the complementary strengths of the dual-encoder architecture for optimal accuracy. Motivated by this observation, our PPGN employs an adaptive fusion strategy to integrate multi-stage predictions.

**Expanding ablation analysis to individual loss terms.** To validate the effectiveness of individual terms, we conduct extensive ablation experiments, with results summarized in Table 9. We define the final supervised loss as $\mathcal{L}_{out} = \mathcal{L}_{ce}(M_{out}, M_q)$, the multi-stage loss as $\mathcal{L}_{ce}^i = \mathcal{L}_{ce}(M_d^i, M_q) + \mathcal{L}_{ce}(M_r^i, M_q)$, and the KL-divergence constraint as $\mathcal{L}_D = \mathcal{D}(\mathcal{N}(\mu, \sigma) \| \mathcal{N}(0, 1))$. As Table 9 shows, the model's performance (mIoU on Chest X-ray datasets)

Table 9: Performance of individual loss terms.

| $\mathcal{L}_{out}$ | $\mathcal{L}_{ce}^1$ | $\mathcal{L}_{ce}^2$ | $\mathcal{L}_{ce}^3$ | $\mathcal{L}_D$ | mIoU (%) |
|---|---|---|---|---|---|
| ✓ | × | × | × | × | 67.13 |
| ✓ | ✓ | × | × | × | 68.61 |
| ✓ | ✓ | ✓ | × | × | 71.32 |
| ✓ | ✓ | ✓ | ✓ | × | 74.28 |
| ✓ | ✓ | ✓ | ✓ | ✓ | **78.14** |

improves steadily from 67.13% to 78.14% when progressively incorporating multi-stage supervision and KL constraints into $\mathcal{L}_{out}$. We attribute this phenomenon to the fact that: (1) $\mathcal{L}_{out}$ ensures global alignment between the final segmentation output and ground-truth masks. It primarily optimizes the fusion parameters $\delta_i$ and $\lambda_i$ in APG, enabling rational weighting of multi-branch predictions; (2) Multi-stage $\mathcal{L}_{ce}^i$ enforces progressive consistency in intermediate predictions (*e.g.*, $M_{r/d}^{1/2/3}$), it directly supervises the iterative refinement in DPG and PRG, guaranteeing reliability across stages — from initial prototype matching to probability-enhanced prototypes and calibrated predictions; (3)

Figure 5: Visualization results of important components. "R" represents ResNet50, "D" denotes DINOv2. "DMPG" and "PRG" mean dynamic mixed-probabilistic prototype generator and dynamic prototype generator, respectively. Areas in red boxes represent incomplete segmentation or background interference.

$\mathcal{L}_D$ regularizes the distribution parameters (*e.g.*, $\mu$ and $\sigma$) of hybrid prototypes generated by MPG, imposing a standard Gaussian prior to prevent over-divergence. This mitigates overfitting caused by limited data (*e.g.*, uncontrolled growth of $\sigma$).

**Visualization results of important components.** Further, we present the visualizations of PPGN's core modules (including DMPG and PRG) in Fig. 5. The results reveal that models employing a single encoder (ResNet50 or DINOv2) achieve only coarse target localization, with segmentation outputs exhibiting substantial noise and/or missing foreground regions (the 2nd and 3rd columns in Fig. 5). In contrast, the combined use of ResNet50 and DINOv2 demonstrates complementary advantages: the CNN-Transformer hybrid architecture simultaneously suppresses background interference and recovers missing foreground details (the 4th column in Fig. 5). With the addition of the DMPG module, the foreground representation is enhanced. We hypothesize that this improvement stems from DMPG's dual-path mechanism (DPG and MPG), which increases the discriminability of query prototypes, thereby enabling more precise target object segmentation. Finally, the PRG module further refines the results through its prediction-prototype-prediction refinement strategy, adaptively fusing multi-stage outputs to boost segmentation accuracy.

**Replacing DINOv2 with SAM and CLIP.** We replace DINOv2 with CLIP and SAM, respectively, to evaluate the model's sensitivity and adaptability under different vision foundation models. The results are shown in Table 10. As Table 10 shows, when using vision encoders based on the ViT architecture, *e.g.*, SAM or CLIP, the model maintains relatively stable performance, *i.e.*, 74.23% and 76.86% mIoU on the Chest X-ray dataset. This further proves that our PPGN exhibits strong adaptability.

Table 10: Compared with different foundational models.

| Method | mIoU (%) |
|---|---|
| CLIP | 74.23 |
| SAM | 76.86 |
| **DINOv2 (Ours)** | **78.14** |

**Complexity analysis.** To further investigate the complexity of the model, we conduct a comprehensive complexity comparison with state-of-the-art models, *e.g.*, GPRN, evaluating total parameters, trainable parameters, inference speed (FPS), and segmentation perfor-

Table 11: Model complexity comparison.

| Method | Total (M) | Learnable (M) | FPS | mIoU (%) |
|---|---|---|---|---|
| GPRN | 99.62 | **9.98** | 10.64 | 66.80 |
| **Ours** | **98.39** | 25.62 | **16.43** | **73.47** |

mance (mIoU). All experiments are performed under identical conditions to ensure fairness, as summarized in Table 11. The results demonstrate that our method achieves superior performance with fewer total parameters (98.39 *vs.* 99.62), along with higher FPS (16.43 *vs.* 10.64), while significantly outperforming GPRN in mIoU on the ISIC2018 dataset (73.47% *vs.* 66.80%). These findings conclusively validate that our PPGN maintains an optimal balance between performance and efficiency. Furthermore, we fully recognize the importance of lightweight architecture design for resource-constrained environments and practical applications. In future work, we will explore more efficient structural designs, such as replacing DINOv2 with MobileSAM, to significantly reduce computational overhead.

