# OpenReview forum: "Probabilistic Prototype Generation Network for Cross-Domain Few-Shot Semantic Segmentation"
_ICLR.cc/2026/Conference — ICLR 2026 Conference Withdrawn Submission_

### Official Review · Reviewer_E6KV · 2025-10-16

**Soundness:** 2
**Presentation:** 3
**Contribution:** 1
**Rating:** 2
**Confidence:** 5

**Summary:**

This paper incorporates DINO as an additional vision encoder for the CD-FSS task, along with several technical designs. While the method achieves strong performance, the novelty, motivation, and computational cost considerations fall short of the acceptance standard.

**Strengths:**

The writing is good and easy to follow.

The perofrmance is good.

**Weaknesses:**

[Major]1-The motivation of the CD-FSS task is inaccurately described. The origin of CD-FSS lies in the fact that we often lack sufficient labeled data in certain specific domains. Therefore, a two-stage learning paradigm is adopted: first learning few-shot capability from a data-rich source domain, and then adapting it to a data-scarce target domain. Researchers introduce the proposal of cross-domain setting is merely to increase task difficulty. This suggests that the authors may lack sufficient background knowledge of this research area.

[Major]2-The higher performance brought by DINO is not surprising. However, the inclusion of DINO significantly increases computational and memory costs, which is a major weakness. Moreover, the authors claim that DINO features are global features without providing any in-depth analysis or empirical evidence.

[Major]3-The overall design lacks novelty. The main framework of this paper merely leverages two encoders and fuses their features, without introducing substantial new ideas or insights.

4-The statement in lines 253–269 is confusing. It is unclear whether the high-confidence area refers to the intersection between the two encoders’ outputs, or if there are two separate confidence maps — one generated from ResNet and the other from DINO.

5-Why do you assume that the prototype follows a Gaussian distribution? The role of using N sample prototypes is also unclear.

6-Authors should discuss the time consumption for their multi-turn predictions.

**Questions:**

Please see the weakness.

---

### Official Review · Reviewer_vjK4 · 2025-10-26

**Soundness:** 2
**Presentation:** 3
**Contribution:** 3
**Rating:** 6
**Confidence:** 4

**Summary:**

This paper proposes the Probabilistic Prototype Generation Network (PPGN) for cross-domain few-shot semantic segmentation (CD-FSS). PPGN adopts a dual-encoder architecture that integrates CNN local features with DINOv2 global priors. To handle the intra-class variation and domain shift in CD-FSS, a Dynamic Prototype Generator (DPG) produces query-specific prototypes from high-confidence response maps, while a Mixed-Probabilistic Prototype Generator (MPG) enhances prototype generalization through probabilistic fusion. An Adaptive Prediction Aggregator (APG) further refines the final segmentation. Experiments show that PPGN achieves state-of-the-art performance on four CD-FSS benchmarks.

**Strengths:**

- This paper is well-organized and easy to follow.
- The proposed framework effectively leverages the strengths of CNNs and ViTs by integrating CNN-based local feature extraction with DINOv2-derived global contextual priors within a dual-encoder architecture.
- Extensive experiments on four CD-FSS datasets demonstrate that the proposed method achieves state-of-the-art performance and validates its effectiveness.

**Weaknesses:**

- The rationale for how probabilistic modeling improves the generalization ability of the prototypes is not clearly explained.
- The paper does not provide a clear justification for choosing DINO over other ViT-based foundation models.
- The ablation study is limited to a single Chest X-ray dataset.

**Questions:**

- The rationale behind how probabilistic modeling improves the generalization ability of the prototypes is not clearly explained.
- Ablation study results corresponding to Tables 2 and 3 should be reported across all four datasets.
- It is unclear whether all ViT-based foundation models would be effective. A clearer explanation is needed for the choice of DINO over other ViT-based foundation models.
- For Table 10, the comparison among CLIP, SAM, and DINO should include results on all four datasets.

---

### Official Review · Reviewer_S2Fy · 2025-10-31

**Soundness:** 2
**Presentation:** 2
**Contribution:** 2
**Rating:** 2
**Confidence:** 5

**Summary:**

The paper proposes a crossdomain fewshot semantic segmentation (CDFSS) method based on the Probabilistic Prototype Generation Network (PPGN). This method fuses local features from Convolutional Neural Networks (CNNs) with the global prior knowledge of DINOv2, integrating the global contextual priors of vision foundation models with probabilistic modeling to generate more discriminative and robust prototype representations. The authors attempt to address the domain shift problem through dynamic and probabilistic prototypes. However, the study exhibits significant flaws in methodology, presentation, and contributions, which are elaborated in detail below.

**Strengths:**

Leveraging the pretrained DINOv2 model to extract global features contributes to improving the model’s crossdomain transferability, leading to certain performance gains on CDFSS benchmarks.

**Weaknesses:**

1) The architecture heavily relies on pre-trained models (ResNet50, DINOv2). Core components such as the Dynamic Prototype Generator (DPG) and Mixed Probabilistic Prototype Generator (MPG) adopt common ideas from existing literature without significant innovation. The probabilistic modeling in MPG lacks theoretical justification and task-specific adaptation. In addition, the Adaptive Prediction Aggregator (APG) module is similar to existing iterative finetuning approaches, offering little novel insight.

2)Ablation experiments are conducted only on the Chest X-ray dataset, failing to demonstrate the cross-domain effectiveness of each module. There is also a lack of rigorous analysis to verify the necessity of the probabilistic mechanisms.

3)Figure 2 is overly complex and fails to clearly illustrate the model's workflow. The paper does not sufficiently explain the interaction mechanisms between modules, particularly the non-deterministic modeling in MPG.

4)The integration of CNN and DINOv2 is simplistic. The probabilistic modeling draws heavily from Bayesian deep learning but lacks meaningful design for the segmentation task. Overall, the work reorganizes existing concepts with minor adjustments, raising doubts about its contribution.

**Questions:**

My main concerns are the limited novelty of the presented work

---

### Official Review · Reviewer_vyat · 2025-10-31

**Soundness:** 3
**Presentation:** 3
**Contribution:** 2
**Rating:** 4
**Confidence:** 5

**Summary:**

This paper introduces a Probabilistic Prototype Generation Network (PPGN) designed for cross-domain few-shot semantic segmentation (CD-FSS). The method integrates both local feature extraction from CNNs and the global contextual priors from DINOv2 to enhance representation transferability. It includes a Dynamic Prototype Generator (DPG) to produce more discriminative query prototypes and a Mixed Probabilistic Generator (MPG) to strengthen generalization through probabilistic modeling. An Adaptive Prediction Aggregator (APG) refines predictions across stages. Experiments demonstrate PPGN achieves state-of-the-art performance.

**Strengths:**

1. The paper is clearly written and well organized, make it easy to follow.
2. The experiments are extensive and well support the proposed approach.

**Weaknesses:**

The motivation of this paper is not very compelling, and its novelty is modest.
1. The paper highlights “incorporating CNN’s local feature extraction with the global contextual priors of DINOv2” as one of its main contributions (L108–L111). However, as shown in Table 2, the overall performance is still primarily driven by the CNN branch, with DINOv2 contributing only marginal improvements.
2. The DPG and APG modules are relatively straightforward and do not demonstrate strong novelty. The only component that seems to provide real insight is the MPG module, as reflected in the paper’s title. However, I have reservations about whether the MPG module truly brings meaningful benefits; see Questions section for details.

**Questions:**

1. After reading the ablation results in Table 3 and Section A.4 of the supplementary material, I still find certain aspects of the MPG module unclear. Specifically, I am unsure why two 1×1 convolutions are required to learn μ and σ. Given the complexity of the overall network and the training losses, I wonder whether these two simple 1×1 convolutions can truly learn meaningful representations, even with the KL-loss constraint—would the model tend to “cheat” or take a shortcut? Furthermore, since the uncertainty U is used to modulate Pmix, this acts as a scalar weighting(E.g.^Pmix = 1.2Pmix or 1.8Pmix). However, as ^Pmix is later calculated with fq through cosine similarity, which involves normalization, I am not sure whether this scalar would still have any actual impact on the final result.
2. Why are the results on the Chest X-ray dataset in Table 1 relatively low—even inferior to some methods without a fine-tuning phase—while the ablation study is conducted based on it? Are the proposed modules also effective on other cross-domain settings or datasets?

---

### Note · Authors · 2025-11-21

I have read and agree with the venue's withdrawal policy on behalf of myself and my co-authors.